# Family Farming and Social and Solidarity Economy Enterprises in the Amazon: Opportunities for Sustainable Development

**Pedro Henrique Mariosa** [1,*] **, Henrique dos Santos Pereira** [2] **, Duarcides Ferreira Mariosa** [3] **,**
**Orandi Mina Falsarella** [3] **, Diego de Melo Conti** [3] **and Samuel Carvalho De Benedicto** [3]

1    Institute of Nature and Culture, Federal University of Amazonas, Manaus 69067-005, Brazil
2    Center for Environmental Sciences, Federal University of Amazonas, Manaus 69067-005, Brazil
3    Center for Economics and Administration, Postgraduate Program in Sustainability,
     Pontifical Catholic University of Campinas, Campinas 13087-571, Brazil
*    Correspondence: pedromariosa@ufam.edu.br

**Abstract:** A research gap in the scientific literature has arisen concerning the challenge faced by actors who formulate public policies on the compatibility between economic activities and sustainable development in the Amazon. The main question that guided this study was whether the organizations of the social and solidarity economy (SSE), in the form of family farming cooperatives and associations, are sufficiently consolidated in the Brazilian Legal Amazon. To achieve the intended objective, the authors used the ArcGIS Pro 10.8 software with an exploratory analysis of spatial data (AEDE). Specifically, the mapping clusters tool was used to present and discuss the distribution of establishments and enterprises in a municipality. The database was the "2017 Agricultural Census" from the Brazilian Institute of Geography and Statistics (IBGE), the most recent official government data available. Establishments and enterprises of family farming in the 772 municipalities of the Legal Amazon with credit access in a period between August 2017 and February 2019 were selected for analysis. We confirmed the hypothesis that SSE projects are essential to achieve sustainability in the Amazon. In addition, this study suggests that this model can be an essential alternative to support public policies for the sustainable development of the biome.

**Keywords:** Amazon; sustainable development; public policies; social and solidarity economy; family farming

## 1. Introduction

The Amazon is a significant, paradigmatic, and provocative locus for the debate about the involvement of different organizations, agents, and actors in the formulation of public policies and in the promotion of sustainability [1]. The dynamic balance between the forces of the market, the state, and society is most apparent in the institutional framework that regulates the relationship between the environmental, economic, and social dimensions of productive activity. In the Amazon's territory, the local impacts are particularly serious, but there are unprecedented opportunities to achieve environmental conservation and the well-being of the planet, people, and different sectors of society, as listed in the Sustainable Development Goals (SDGs) provided by the United Nations 2030 Agenda [2,3].

The sustainability advocated for the Amazon region requires, therefore, a plural and multidimensional approach. It incorporates reflections from the ecological approach in that it insists on supporting the recovery and reproduction of ecosystems (resilience) in the face of anthropic or natural aggressions [4]. It also considers the assumptions of the economy to be essential, with the term sustainable development often being used to discuss this relationship [5].

Sustainable development, by this definition, applies to the set of actions conducted to improve the living conditions of the population, within the limits of the sustaining

capacity of ecosystems, by integrating their social, economic, environmental, political, and cultural dimensions [6,7]. The economic dimension is related to the creation, accumulation, and distribution of wealth; the social and cultural denotes quality of life, equity, and social integration; the environmental refers to natural resources and the sustainability of medium- and long-term models; and the political dimension deals with aspects related to territorial governance, as well as the sustainable collective project. Finally, sustainable development is about apprehending territories in the form of or as networks of interaction and collaboration [8,9].

Sustainable and solidary development has been conceived of as an alternative to the current dominant pattern of development that promotes environmental degradation and social insecurity [10]. Essentially, this model seeks economic production that coexists in harmony and in tune with the cycles of nature, enough to obtain not only subsistence, but also the construction of new paradigms for obtaining income and productive inclusion without conventional aggressions to the environment, intentional or not.

Family farming accounts for over 90% of all farms in the world and produces 80% of the world's food by value. It is the main driver of sustainable development, including ending hunger and all forms of malnutrition. Public policies aimed at supporting family farming aim to create an enabling environment that strengthens its position and maximizes its contributions to global food and nutrition security [11]. In this study, we first examined the presence and distribution of agricultural activities in the Brazilian Amazon, considering and comparing the two ways in which the establishments are structured, family and non-family farming; we then considered the presence and contribution of social and solidarity economy enterprises in the formation of interaction and collaboration networks for local and sustainable territorial development. Therefore, its academic contribution and its ability to serve as a basis for the promotion of public policies aimed at cooperatives and associations of family farming in Brazil are given as justifications for the elaboration of this study.

Federal Brazilian Law no. 11326 of 2006 defines what characterizes family farming in Brazil. To be classified as a family farm, the establishment must be small (up to four fiscal modules) and have strictly family management, with at least 50% of the workforce supported by family members whose income is acquired through activities carried out in the establishment. In contrast, non-family farming encompasses all other establishments not covered by Law no. 11326 [12]. Non-family farming groups also include small- and medium-sized establishments that do not fall under the law due to area limits or income limits, as well as those that are public lands [13].

The social and solidarity economy presents itself, therefore, as a form of economic organization that, without letting go of the basic precepts of productive activities regulated by market laws, emphasizes social and human demands and requirements [14–16]. Social and solidarity economy enterprises (SSE) are important elements for achieving the sustainability of the Amazon, since they aim to promote the organization and consolidation of territorial systems of sustainable economic development through providing socio-environmental and economic integration for the population living in this region [17,18]. Family farmers organize themselves into self-help groups and cooperatives. By organizing themselves economically into agricultural cooperatives and politically into associations, they become important social actors in the dialogue and defense of public policies for sustainable development. As SSE enterprises, family farms can address both market failures and state failures (namely, the neglect of agriculture in recent decades). In addition, family farmers prefer the use of agroecological production methods with low inputs and low carbon emissions, and which respect the principles and practices of agrobiodiversity conservation. Alternative food networks associated with fair trade, joint purchase, and collective supply complement the role that family farmers' joint ventures can play in promoting more sustainable agri-food systems [19].

In this perspective, the main question that guided this study was whether SSE enterprises, in the form of cooperatives, associations, and local collectives of family farming,

were sufficiently developed in the Brazilian Legal Amazon and, territorially, in what way. In response to this question, the authors aimed to analyze the spatial distribution of establishments and enterprises of family farming to determine which principles, values, techniques, and aims of territorial development are present in the Legal Amazon, which, through comparisons with establishments of non-family farming, may be taken as characterizing the SSE.

*1.1. Background*

1.1.1. The Concept of a More Solidary Economy

Under the name of "economic systems", countries, on the initiative of society itself or its rulers, establish the guidelines, rules, and norms by which goods and services will be produced, distributed, and consumed [20]. Therefore, the forms of organization of economic activities can define how existing wealth should circulate, who and which sectors will receive help from the financial and economic resources amassed, and how and in what way the distribution of these resources will or will not distinguish the members of a certain social stratum [21]. Although they have different gradations and possibilities, two polar types of economic systems have been organized and structured historically. On the one side, there are those in which there is greater state intervention and, on the other, there are liberal economies more open to individual initiatives, in which the presence of the state tends to be more reduced or discouraged [22].

Notwithstanding the arguments of its defenders, there is no record of the functioning of any economic system without a minimal institutional framework and without a minimum of juridical-legal rules or forms of control. Likewise, there is no historical record of measures able to predict and regulate all the nuances and details of how economies should work [23]. What was sought at the most distinct socio-political moments in history in historical-social configurations was a legitimately recognized and accepted way to know who decides what to produce and in what quantity, as well as how, by whom, and for whom they will be produced [24].

In a so-called market economy, based on the private ownership of the means of production, which provides for the free initiative of entrepreneurs and the predominance of competition between the various producers, the goal of its agents is to achieve rewards for the capital invested in its operation. In other words, shareholder profit is the incentive for the economy to self-regulate and achieve the necessary balance between those who produce and those who consume, under the relentless laws of supply and demand [25]. The main criticism of this model, however, is that what we have, in fact, is the overvaluation of profit and the emphasis on results for investors to the detriment of any other interests that may imply a reduction in the gains to be obtained from economic activity. In these economies, state intervention is strongly rejected, except to protect private property and the freedom of economic agents [26].

In a centrally planned or state economy, the exact opposite occurs. In countries with a socialist political regime, decisions about what, how, and for whom to produce are made at the state level by a central planning body. In this sphere, and not in the market via prices and competition, there is a coordinated effort to try to adjust needs to available resources, defining the tasks, duties, and obligations of each economic entity [27]. The main criticism of this model focuses on bureaucratic rigidity and the excessive control of economic activities, which encourages corruption and the inefficiency of supply systems, drastically reducing dynamism, innovation, and risk incentives, which are so salutary in market economies [28].

In a mixed economy, in turn, the state and the market complement each other. As, in fact, there is no record of a society in which the economy totally follows the market model or is totally planned, all economies are intermediate between these two models. In mixed economies, we find a balance between public and private companies, depending on the degree of state intervention in the economy. This intervention, however, is not restricted to the ownership of companies or shareholders' participation in businesses; it also con-

cerns other instruments, such as legislation, taxation, government procurement programs, subsidized credit, and the promotion of activities considered politically strategic [29].

Whatever the principle or foundation that justifies the existence of economies, be it: for market economies, capitalist private appropriation; for state economies, collective appropriation; or, for mixed economies, the participation and strong influence of the state in the results of private capitalism, its main elements are the prioritization of profit and the subordination of work to the demands of capital. These create, however, gaps, incompleteness, and distortions that take shape in the dimensions of life and human activity, which are disregarded, or simply not seen, when viewed from the exclusive angle of the economy [30,31].

In this context, aiming at a greater balance between economic and socio-environmental development, the social and solidarity economy presents itself as an alternative to the current development model [32,33]. The social and solidarity economy expresses the idea of many, varied forms of economic organization, such as the production of goods and merchandise, the provision of services, marketing systems, financial operations, and consumption, which have in common the fact that they are based on work, are associated or cooperative, are commanded and directed in self-management regimes, have their resources managed in the form of the collective ownership of the means of production, and have as guiding values cooperation, sustainability, and solidarity [34]. Concretely, it is possible to affirm that, though there are other possibilities, its main organizational characterization is a cooperativism or associativism structured in the format of solidary and sustainable economic enterprises.

It is worth noting that one of the main strategic goals of social and solidarity economy enterprises is to promote the organization and consolidation of territorial systems for sustainable economic development [35,36]. In these, the main parameters for the actions of the enterprises, regardless of the support or lack thereof of specific public policies, are the economic, social, and environmental conformations of the territories they cover [37]. With territorial development actions, SSE enterprises seek socioeconomic and cultural mechanisms that contribute to the inclusion of vulnerable populations and to people continuing to live in the places they were born, which is a priority for young people [38].

In sustainable and solidary development, the potential endogenous production systems are valued based on social technologies suitable for local contexts, which is why it favors the preservation of the values of traditional peoples and communities. In addition, economic activity must be integrated with the support of the environment in which it is being conducted [39].

In this sense, the social and solidarity economy seeks to project itself as a paradigm and model of development that is based on an economic model that values cooperativism, participation, inclusion, and self-management for community development, articulating the preservation of nature through sustainable management [40].

### 1.1.2. Policies to Strengthen Family Farming

PRONAF (National Program for the Strengthening of Family Farming), created in Brazil in 1995, became an important state mechanism in the 2000s for the legitimation and recognition of family farmers as a social category [41]. Bringing together in a category those that were previously designated by terms such as small producers, family producers, low-income producers, and subsistence farmers, the program also made it possible, in the form of public policy, to strengthen this sector with four main lines of action: (a) Credit costing and investment; (b) Financing of infrastructure and services to municipalities; (c) Training and professionalization of family farmers; and (d) Financing of research and rural extension, which all aim at the generation of technologies and their transfer to family farmers.

The program also has started to intensify the sources of financing for family farming, e.g., agencies such as BNDES, the Worker Support Fund (FAT), Constitutional Funds of the Northeast (FNE) and of the Midwest (FCO), and cooperative banks, which work through

agreements with Banco do Brasil [42]. In addition to the federal banks, Banco do Nordeste do Brasil-BNB and Banco da Amazônia Sociedade Anônima-BASA are also involved.

The PRONAF guidelines also show clear parameters within the PRONAF Aptitude Document (DAP) for the impact of this intersection at the national level. The issuance of the DAP presupposes a specific classification for associative forms of family farming and family rural enterprises, as well as an overview of the contribution of resources to this specific niche [43].

The DAP classifies family farmers into four groups ("A", "B", "A/C", and "V"). The "B" group consists of family farmers with an annual family income of up to BRL 23.000,00. The "V" group consists of those with an annual family income of up to BRL 415.000,00. The other two classifications refer to specific groups of beneficiaries of public policies in relation to rural settlements.

The issuance of the DAP allows family farmers access to fifteen different public policies: 1. Technical Assistance and Rural Extension (ATER); 2. Family Farming Insurance (SEAF); 3. Warranty-Harvest; 4. Minimum Price Guarantee Program (PGPM); 5. Family Farming Price Guarantee Program (PGPAF); 6. Food Acquisition Program (PAA); 7. National School Feeding Program (PNAE); 8. National Biodiesel Use and Protection Program (PNPB); 9. Special Beneficiary of Social Security; 10. Rural Retirement (FUNRURAL); 11. Emergency Financial Aid; 12. Minha Casa Minha Vida Rural Program; 13. Brazil Without Extreme Poverty Plan—Rural Productive Inclusion Route; 14. Quotas in Vocational Schools (CEFET); and 15. Pronatec Field [44].

Ordinance No. 523 of 24 August 2018 defines the associative legal forms of family farmers and rural family enterprises and appoints four modalities of these collective organizations, which, in Article 9, are presented with their respective identification parameters:

I—Rural Family Enterprises: constituted for the purpose of processing and marketing agricultural products, and even for the provision of rural tourism services, provided that they are constituted exclusively by one or more family farmers benefiting from the UFPA DAP (Family Unit of Agricultural Production); II—Single Family Farming Cooperatives: made up of at least sixty percent of cooperative family farmers benefiting from the UFPA DAP; III—Central Cooperatives of Family Farming: constituted exclusively by individual cooperatives associated with DAP Legal Entities; IV—Family Farming Associations: made up entirely of members benefiting from DAP Legal Entities and with at least sixty percent of the individual members benefiting from DAP or demonstrating both situations in the case of mixed composition [45].

In this study, we considered that the field of the social and solidarity economy includes in its framework all enterprises and collective community organizations, such as enterprises, cooperatives, and workers' associations. In this sense, the four legal forms of family farming associations mentioned above are linked to the field of SSE enterprises.

We also considered that public policies for sustainable territorial development only become effective if formulated based on collaborative processes and provided with complete and updated information on where, for whom, and with what goals they were or will be formulated [46,47]. Sustainable territorial development can be understood as a philosophy of participatory planning and management that considers the distinctive characteristics and heritage of a given territory [48].

Consequently, the social and solidarity economy promotes, in family farming, and specifically in extractive communities in the Amazon, social transformations through the generation of work and income, the restructuring of the activities and practices of societies, and social inclusion in the contexts of cooperation and solidarity [49]. It is through this perspective that social transformations are developed by the communities, through the elaboration of products, methodologies, or techniques, that is, by their taking over of social technologies and incorporating them into their communities. Those social transformations implemented within the scope of partnerships between civil society organizations and government entities have a significant ability for dissemination, as these organizations

complement each other's needs for aid and regulation, which is the basis for the production and replication of social technologies [50].

Regarding the background of the two themes, there is in a previous study [51] a more in-depth analysis of the two themes, through a systematic review of the literature on family farming and the social and solidarity economy in Latin America.

## 2. Materials and Methods

Regarding the study design, to achieve the goal proposed in this study, a methodological approach was adopted based on the analysis of the spatial distribution and autocorrelation of family farming establishments and enterprises in the Brazilian Legal Amazon. It was an exploratory study of a spatial nature and gathered data from documentary and bibliographic sources [52,53].

In the data collection process, we considered family farming establishments as any production or exploration unit dedicated, totally or partially, to agricultural, forestry and aquaculture activities [54]. The main database used was the Agricultural Census, launched on 25 October 2019, and namely the "2017 Agricultural Census", which, among various contributions, makes a set of query tables available on the SIDRA platform of the Brazilian Institute of Geography and Statistics (IBGE).

In order to identify family and non-family farming establishments for their subsequent distribution in municipalities of the states of the Legal Amazon, public access databases were used, such as: http://dados.gov.br, accessed on 21 August 2021, particularly the data set SEAD—DAP Actives of Legal Entities 3-2 with SSE enterprises classified as those that appeared at least once with active DAPs from August 2017 to February 2019, which is the most recent official government data available. The main database, with information on data from enterprises with active DAPs, was organized and stratified according to the type of DAP (family farming cooperative, family farming association, or family rural enterprise), the absolute number of members, number of partners, individual and legal entity, state (UF), and municipality.

Regarding the coverage area, the Brazilian Legal Amazon (Figure 1) encompasses around 5 million km$^2$, covering about 59% of Brazilian territory. Its space currently includes 772 municipalities distributed in the seven states in the north region, in addition to Maranhão (the state in northeast region) and Mato Grosso (the state in midwest region). Among these, Acre has 22 municipalities, Amapá 16, Amazonas 62, Maranhão 181, Mato Grosso 141, Pará 144, Rondônia 52, Roraima 15, and Tocantins 139 [55].

With a low demographic index, the area only contains 12.3% of the entire Brazilian population. One should also note that of the 5 million km$^2$, the north region contains about 3.8 million km$^2$, of which 76.28% is "essentially rural". Another 20.05% is "relatively rural", leaving only 3.66% of the occupancy area "essentially urban" [56].

As a unit of analysis, we considered the aggregated establishments in the form of social and solidarity economy enterprises. An establishment denotes any production or exploration unit dedicated, totally or partially, to agricultural, forestry, and aquaculture activities [57]. Enterprises are all collective organizations linked to the production, processing, and marketing of products linked to agrobiodiversity, such as cooperatives and other forms of collective and/or social enterprises, informal groups, community organizations, and associations of workers in the formal and informal economy [14,58].

Regarding the data criteria, we defined the eligibility criteria of the enterprises and establishments as follows:

(a) Inclusion: we selected for further analysis establishments and enterprises of family farming in the 772 municipalities of the Legal Amazon with active DAPs in the period between August 2017 and February 2019.

(b) Exclusion: we excluded all non-registered third sector organizations with active DAPs, public companies, and employers, including family rural enterprises composed of only one partner.

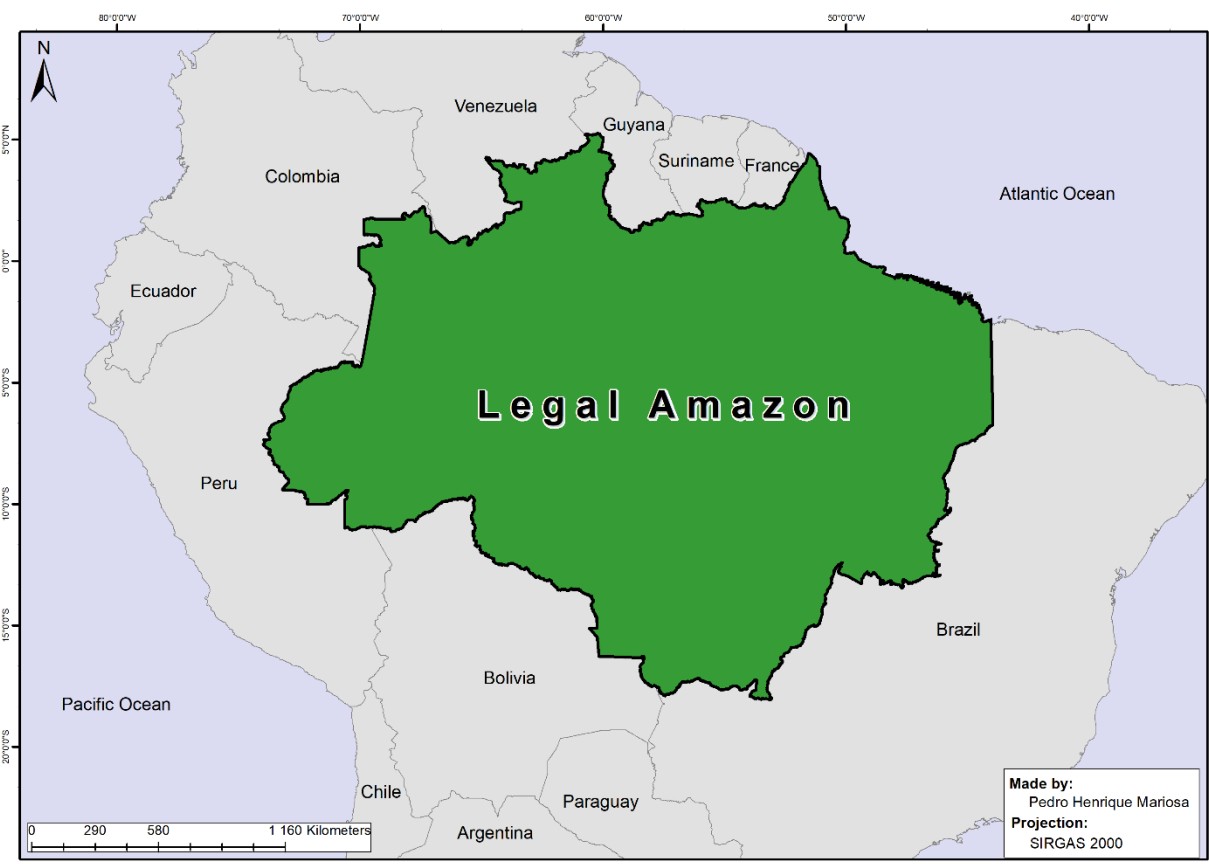

**Figure 1.** Brazilian Legal Amazon Map.

Of the enterprises selected in this first review, we gathered information in the main database, considering only those with active DAPs, and then organized this according to the type of DAP (family farming cooperative, agriculture association or family rural enterprise), corporate name, branch of activity, and location (municipality and state).

The statistical analysis of the data sought, first, to structure the information and stratify patterns of dispersion of enterprises by geolocation. We applied multiple linear regression models [59] to estimate the representativeness of the SSE enterprises of the municipalities of the Legal Amazon, crossing information from the points of concentration of the enterprises. We assessed their representativeness considering two variables: (a) the total number of family farming enterprises of the social and solidarity economy in each municipality; and (b) the number of non-family establishments and family establishments in each of the municipalities.

For the interpretation of data, we applied a spatial model to look for statistical evidence of internal association between the two variables found and collected. For this purpose, we used the spatial analysis technique provided by the Moran Local Index, performed with the aid of the ArcGis software [60]. For the analysis of spatially distributed data, this approach attempts to find the degree of spatial autocorrelation of the observed variable and the neighboring sample units. In general terms, it is about deciding whether there is no spatial autocorrelation ($H_0$ or null hypothesis), or if spatial autocorrelation ($H_1$ or alternative hypothesis) can be proved.

It so happens that the Moran Index provides us with the general measure of spatial association or autocorrelation present in the global set of analyzed data. Values close to (0) show non-existence; positive values (1) point to the existence of spatial autocorrelation; negative values (−1) show the absence of spatial autocorrelation and are valid for the entire area.

We calculated Moran's coefficient I (Moran I) for each distance class (*d*), according to the following equation [56]:

$$I(d) = \left[\frac{n}{Wd}\right] \frac{\sum ni = 1 \sum nj = 1 wij(d)(xi - x^-)(xj - x^-)}{\sum ni = 1(xi - x^-)2} \qquad (1)$$

where:

- $n$ is the number of units;
- $wij(d)$ is the connectivity matrix of the distance class $d$ (also known as the weight matrix);
- $Wd$ is the sum of all $wij(d)$, being the number of pairs of locations per distance class;
- $xi$ and $xj$ are the values of the variable of interest in (local) units $i$ and $j$.

When, however, it was desired to analyze subsets of data that were separated by any distance, the Moran index was applied locally.

Luc Anselin proposed the Local Moran Index (Local Indicators of Spatial Association—LISA) as an auxiliary tool in the analysis of spatial aspects of data that is capable of testing local autocorrelation and detecting spatial objects with greater or lesser influence on the global Moran index. The statistical approach analyzes the covariance found in the different area units. In the first case, the Global Moran Index detects the spatial autocorrelation between all the polygons seen in the study. In the second case, the Moran Local Index shows the existence and level of autocorrelation between a polygon of interest and its neighborhood, defined from a distance (d) [61]. In this way, it is possible to use Moran's Local Index as a statistical tool to find the existence and extent of significance of a "cluster" of equal values. In the interpretation of Moran's Local Index, significantly higher and positive values allow for inferring the presence of a "cluster" or grouping, configuring it in the form of equal, high, or low values. Already significantly low values point to a situation of inequality in the analyzed region, such as an "anti-cluster" pattern, which is indicative of transition zones between one form of spatial autocorrelation and another.

Regarding the analysis and interpretation of data, the establishments found in this study were distributed in spreadsheets as attributes of the municipalities. With each municipality named by its spatial location in x and y geographic coordinates, the ArcGis program performed the conversion of data from geospatial points to area data or polygons, in which we applied the Moran local index.

The elaboration of this type of polygon has the fundamental principle that "considering a territory, there are points that are closer to a generating source than to another source, and the result is a polygon whose distances between the source and the point are the smallest possible" [62]. This diagram has been used in different studies and applications, such as the analyses of influence areas and services in schools, stores, and hospitals [63–65].

After the spatial distribution of establishments and enterprises by municipality was established via an exploratory spatial data analysis (ESDA) using ArcGis Pro 10.8 software [66], it was possible to describe the spatial distribution and understand the patterns of spatial association. These were synthesized in three maps with municipalized information on non-family and family farming and the number of SSE enterprises by municipalities in the Legal Amazon.

Then, also using the ArcMap 10.8 software, we applied the mapping clusters tool to find patterns of spatial association, using spatial autocorrelation [67]. The mapping clusters tool used input data, such as establishment numbers per municipality in an output resource, finding them on the map in clusters and outliers, which were formed from three sets of statistical data, the Moran Local Index, z-scores, and $p$-values.

Z-scores and $p$-values are measures of statistical significance that suggest, feature by feature, whether the researcher should reject the null hypothesis. They show whether there was a clear similarity between a spatial cluster (called a cluster) or dissimilarity (called an outlier). A high positive z-score for a municipality shows that the surrounding municipalities had similar values (high values or low values), being found on the map as HH (high–high) for a statistically significant cluster of high values, and LL (low–low) for a statistically significant cluster of low values. A low negative z-score for a municipality shows a statistically significant spatial data outlier and is found on the map as HL (high–low) for a municipality with a high value and surrounded by municipalities with low values,

or LH (low–high) for a municipality with a low value that is surrounded by municipalities with high values. In the cluster map, statistically significant clusters and outliers are always shown for a confidence level of 95%. When there was no statistical significance, clusters and outliers are shown on the map as "not significant" [66–68].

With the mentioned statistical procedures, it was possible to estimate the representation of SSE enterprises of family farms in the Legal Amazon.

## 3. Results

Several previous researchers have discussed the complementary, conflicting, and antagonistic elements of farming activity in Brazil. In the period from 1970 to 2006, the development of agriculture in Brazil followed a continuous growth trajectory, due to accumulated productivity gains. In this regard, it is suggested that "while the output of agriculture—a combination of vegetable production, livestock, and rural agroindustry—grew by 243% between 1970 and 2006, the use of inputs grew by only 53%" [69]. Other factors besides the use of inputs also contributed to this gain in productivity, such as the qualifications of the workforce, the growing mechanization of processes incorporating information technologies, the opening of new planting fronts, and investment in research. However, previous researchers reiterate that this did not occur homogeneously and with the same rates in all states and regions of the country [70].

In fact, what we consider and analyze from the perspective of Brazilian rural space is highly diversified and multidimensional. It is a sociocultural space that, while encompassing "gatherers and other landless farmers, traditional farming systems of Indigenous peoples and other traditional peoples and communities, family and non-family farming", also constitutes an economic space in which production is "intended from own consumption or sale to establishments exclusively dedicated to marketing and export, guided by the logic of agribusiness and large properties" [71].

Further, the economic agents circumscribed in "family farming, or small farming as it has already been called, differ from the large rural entrepreneur, not by size, but by the social values and by the social, economic, and political logic that guides them, which is another" [72]. As seen in the document "Atlas of Brazilian rural space" [69], multispatiality and multi-territoriality are aspects through which the "family articulates scales and generations in a dynamic and complex network of relationships, not limited to the family nucleus restricted to the property".

This multi-territoriality consists of the social relationship between territory zones (strictly on a spatial scale) and connection networks (beyond the spatial scale), with this combination being the basis of the adaptation strategy used by families belonging to communities and traditional peoples [73,74].

Consequently, family farming is "a productive system that articulates different temporalities and different spatial patterns, and which allows the social reproduction of the family in the countryside or in the city, not only in economic terms, but also in cultural terms". It is in this way that family farming differs from the practices of rural entrepreneurs and non-family farming; this "transmission of memories and cultural, material or symbolic practices, gives the family farming internal coherence and the ability to develop specific strategies of resistance against hegemonic economic dynamics". Turning to the dimensions of sustainability, the document produced by the IBGE [69] points out that the "maintenance and reproduction of family knowledge also favors the continuity of agricultural practices that are more harmonious with the environment, bequeathing to family farming a significant role in environmental preservation, even in the most modernized production establishments".

Let us compare, therefore, the two types of farming: non-family and family farming.

### 3.1. Non-Family Farming and the Agricultural Frontier in the Brazilian Legal Amazon

The Legal Amazon, in its 772 municipalities, was found to have 163,373 establishments characterized as non-family farming establishments (Figure 2), representing 13.89% of the

total 1,175,916 non-family farming establishments in Brazil. The most relevant states are in the so-called agricultural frontier, which, according to [56], consists of an expanding agricultural zone characterized by increasing agricultural modernization. The activities that take place in the states found in the midwest region of Brazil and northward are the protagonists of this movement, namely in the Rondônia, Tocantins, and southern Pará states, and parts of the Mato Grosso, Maranhão, and Tocantins states. Consequently, since the beginning of agricultural frontier expansion, there has been fierce pressure on the Amazon rainforest, which has been illegally suppressed, degraded, and fragmented [68].

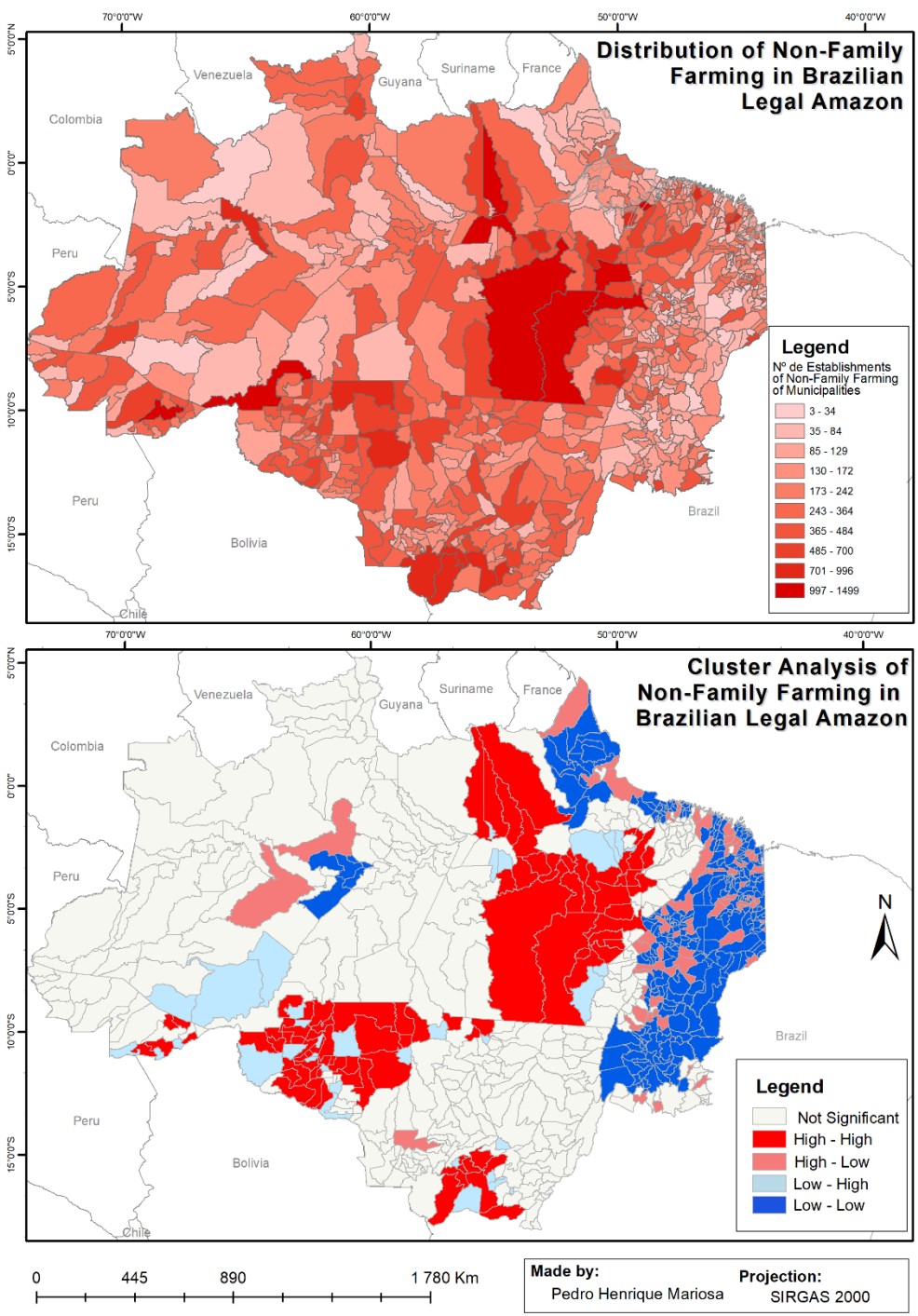

**Figure 2.** Non-family farming establishments' (*n* = 163,373) distribution in Brazilian Legal Amazon.

In the cluster analysis, we found two clustering patterns: a dispersed pattern in the high–high cluster and another more concentrated pattern in the low–low cluster. The

high–high cluster, which included the concentrations of municipalities with many non-family establishments and which was also surrounded by many non-family establishments, presented itself in a significant range from the north to the south of Pará, predominant in all of Pará, the territory of Rondônia, northwest and south of Mato Grosso, and the metropolitan region of Rio Branco, in Acre. These areas coincided with the areas with the highest rate of mechanization, ranging between regular and remarkably high.

Considering the variation in the index [67], to measure agricultural production in the period from 1970 to 2006, we established an initial value of 100 corresponding to the year 1970. We found that in Pará in 2006, this index was already at 328.7; in Rondônia, it reached a surprising 1024.2; in Mato Grosso, the value of the index was 643.6; and in the state of Acre, it reached 266.9. There would not have been such expressive gains in productivity in these areas if it were not for the investments that took place in inputs, machines, processes, qualified labor, and scientific research, which, together with the incorporation of new areas of land, were guided by land concentration, the priority of exports, intensive capital, and the reduction in the number of workers to a minimum.

Pará is the most representative state in the Legal Amazon, recording 41,962 establishments, and an average of 291 establishments per municipality. Pará has a great diversity of agricultural modernization, with a predominance of mineral extractive activities and significant advances in agricultural production on its southern border with Mato Grosso and Tocantins [73].

In Rondônia, the introduction of soybeans in the 1990s in the interior of the state imposed processes of territorial disputes in the face of the expansion of the agricultural frontier [74]. In Rondônia, 17,109 establishments were recorded, and it had the highest average in the Legal Amazon, with 329 establishments per municipality. In this state, the intense process of globalization has manifested itself in a fragmented way; that is, the prevalence of the logic of capital over social relations has been imposed in relation to territorial planning, since the expansion of the agricultural frontier has caused the expulsion of family farmers to other rural areas or to cities [75].

In turn, the pattern found and classified as a low–low cluster consisted of an uninterrupted area covering the south of Tocantins to the north of Amapá, passing through the states of Maranhão and Pará in a similar pattern, and also through the regional metropolitan area of Manaus. This low–low cluster presents a concentration of few establishments in areas with a regular to high mechanization index.

In fact, when the variation in the index of production in this cluster is observed, it can be seen that in Tocantins, for the period between 1995 and 2006, there was an increase of 8.73% in the growth rate of agricultural production; in Amapá between 1970 and 2006, this rate varied by 19.5%; in Maranhão, by 318.4%; and in Amazonas, the rate decreased by 26.6% [67].

The dynamics of land use and land cover in the interior of Tocantins is directly linked to changes in socioeconomic standards, which have undergone transformations resulting from the consolidation of the agricultural frontier expansion zone, which has caused a considerable decrease in areas of native vegetation mainly linked to the Cerrado biome, due to the increase in areas destined for pasture [76].

Maranhão, with 26,291 establishments in the municipalities of the Legal Amazon and an average of 145 establishments per municipality, presents, in addition to the expansion of grains, a recent process of advancement of eucalyptus silviculture on the agricultural frontier of the state. The impacts of large enterprises in the region, aimed at the production of export commodities, have promoted the unrestricted looting of natural resources with little local socioeconomic return [77].

Amapá, which also appeared in the low–low cluster, recorded the lowest total number of establishments, 1523, and the lowest average of 95 establishments per municipality. It has recently become a frontier of agribusiness expansion (soybean, corn, and cotton), and is involved in a broader process of capitalist accumulation on a global scale, mischaracterizing spaces traditionally occupied by gathering. It is under increasing pressure to expand its

soybean planting activity, and the ecological zoning conducted in the state had the sole purpose of increasing agricultural production, without the inclusion of social agents from local family farming and traditional peoples [78].

The agricultural frontier, now also in the Northern Amazon, and expanding its operations in the states of Amapá and Roraima, is the most recent territory for productive investments by national and foreign agri-food corporations. This has raised debates on ethical aspects of the use of transgenic products, as well as on the socio-environmental impacts of the continuous use of pesticides and the increase in deforestation rates.

We saw that in both clusters there was a positive relationship between the number of non-family establishments and the mechanization index. Given this positive relationship, the observation of the evolution of non-family establishments in the new agricultural frontiers in the Northern Amazon shows the need for investments and the inclusion of family farming establishments for the sustainable development, balance, and socio-environmental conservation of these territories.

### 3.2. Family Farming in the Legal Amazon

Consisting of small establishments with family management and workforces, family farming manages the annual movement of BRL 107 billion in the economy, employing more than ten million people. In the 772 municipalities of the Legal Amazon, there are a total of 702,479 (Figure 3) family farming establishments.

The four states with an average of above 1000 establishments per municipality were also those with similar standards. The high–high clusters in conjunction with low–high outliers occurred in the states of: Pará, with an average of 1665 family farming establishments per municipality; Rondônia, with an average of 1429; Acre, with an average of 1414; and Amazonas, with an average of 1414. That is, they were found to form groups of municipalities with high development potential and the presence of family farming establishments, with there being few municipalities in the peripheral area of the cluster with a low concentration of these.

Another distribution pattern found in the study was formed by the four states with an average of below 1000 family establishments, represented by the low–low cluster and high–low outliers. They were the states of Maranhão, with an average of 775 establishments per municipality; Mato Grosso, with an average of 579; Amapá, with an average of 437 and which, despite following a similar pattern, was isolated from the main group; and Tocantins, with an average of 323. That is, in these states, municipalities were found to form groups with a low potential for the development of family farming establishments, with the presence of municipalities with a high concentration of these being sporadic.

It should also be noted that the state of Roraima, which was closer to the general average of the nine states (959 establishments per municipality), was outside a significant distribution pattern, recording an average of 874 establishments per municipality. Finally, in the comparison between the high–high clusters in Figure 2 and the low–low clusters in Figure 3, it should be noted that on the agricultural frontier, especially in the south of the state of Mato Grosso, there was a strong concentration of non-family establishments, and that there was a transition from this position to the low concentration of family establishments to the northeast of Mato-Grosso, the southeast of Pará, and the southwest of Tocantins.

This stems from a model of agricultural development for the region that ends up generating paradoxical situations that, on the one hand, favor an agricultural elite, characterized by the expansion of commercial crops, but on the other hand, generate concerns for family farmers, who despite practicing stable agricultural activity, are threatened in terms of credit limitations and shortages of labor due to aging, in addition to the need to reorient their activity in the face of environmental problems [68].

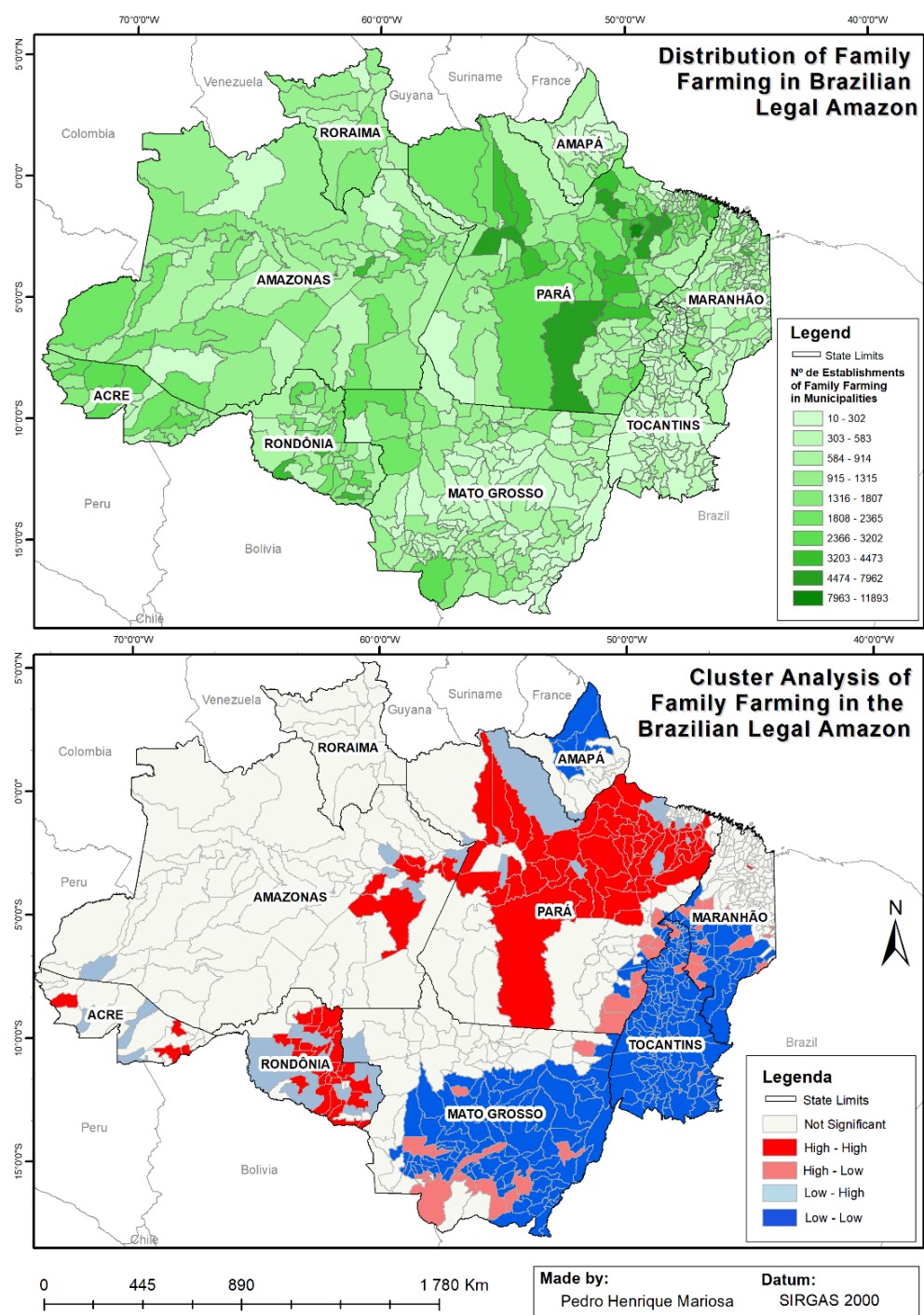

**Figure 3.** Distribution of family farming establishments (*n* = 702,479) in the Brazilian Legal Amazon.

Pará, despite appearing from east to west in its municipalities as a cluster with a strong presence of family farming, is still subject to the policy of development of the agricultural frontier, which has begun to receive the influence of government policies to support soy in regions not only on the edge of the border (Altamira and São Félix do Xingú mainly), but in places such as the Santarém plateau, which has caused territorial disputes, since the organizational and financial power of the so-called sojeiros (soy farmers) has started to be supported by the political project of the state to transform the municipality of Santarém into an agribusiness hub. Therefore, family farming in the territory has been left at a complete and absolute disadvantage in a series of disputes over "working land", a situation that has left communities completely disoriented, not knowing how to deal with the situation [74].

*3.3. Distribution of Family Farming Social and Solidarity Economy Enterprises*

For all agents that integrate the agricultural supply chains, in addition to the demands of consumers and distributors regarding quality, promptness, and safety, it is expected that they meet an always updated set of new requirements, involving guidelines from sanitary and environmental legislation and labor practices and governance, which, in the case of family farming establishments, require knowledge of planning, management, tax, and legal aspects that their members, individually, are not sufficiently prepared to deal with or do not have the financial resources [70]. It should be noted, therefore, that places where a greater number of family farming establishments is concentrated in the form of social and solidarity economy enterprises is a sign that the challenges of implementing environmentally sustainable, socially fair, and economically workable production are being overcome.

For the analysis presented here, it is relevant to highlight that approximately 44% of the municipalities in the Legal Amazon did not even have a cooperative association or a family farming rural enterprise with an active DAP in the analyzed period according to the available data. In Tocantins, for example, 106 municipalities (~76% of the total) were without any representation of these enterprises.

In turn, despite the high population concentration in the capitals of the states of the Legal Amazon, seven out of the nine states (Amazonas, Amapá, Maranhão, Mato Grosso, Pará, Rondônia, and Tocantins) had municipalities with the same or a greater number of SSE enterprises than the capital itself, while the metropolitan regions were, in themselves, potential centers for these types of enterprises.

The availability of resources to finance activities was also unequal. Taking the municipalities in which the family farming SSE enterprises had access to credit, the following distribution was recorded. In Tocantins, they amounted to only 23.74%, and in Mato Grosso, 41.84%. In an intermediate level were the states of Rondônia, with 57.69%, Amapá, with 62.50%, and Maranhão, with 67.96%. The municipalities with a high rate were the states of Pará, with 72.92%, Amazonas, with 75.81%, Acre, with 77.27%, and Roraima, with 80%.

The high–high category refers to the clusters formed by a municipality with a larger number of family farming SSE enterprises surrounded by other municipalities also with many SSE enterprises in this category. With this, it is possible to find the regions of greatest influence on the principles, values, techniques, and models of action typical of the SSE. On the other hand, the low–low category refers to clusters formed by a municipality with few SSE enterprises and that is close to other municipalities that also have a small number of these types of enterprises. We suggest, in this case, the low influence of the models proposed by the family farming SSE enterprises.

In the cluster mapping (Figure 4), then, five patterns were observed. Of the five clusters, two were in the state of Amazonas. The first was in the centre/southeast region of Amazonas, which, in addition to the capital, showed nearby municipalities, such as Itacoatiara, Presidente Figueiredo, and Manacapuru, and some connections with the municipalities of Beruri, Borba, and Novo Aripuanã as crucial points.

The second cluster, between Acre and Amazonas, included, in addition to the capital Rio Branco, most municipalities in the east of the state of Acre, in addition to Lábrea and Canutama in Amazonas. Note that despite not forming a cluster, both Porto Velho in Rondônia and Manicoré in Amazonas recorded a high number of SSE enterprises, and that together with the municipality of Humaitá, they formed an important link between these two clusters.

Closing the Western Legal Amazon group was the third cluster, a pivotal point that brought together municipalities around the capital of the state of Roraima, Boa Vista, and on the axis of the land connection with Presidente Figueiredo and, later, to the Amazon capital, Manaus.

The fourth cluster, forming the Eastern Legal Amazon group, was between Amapá and Pará, and this had a pivotal point in municipalities on the border, such as Mazagão, Pedra Branca do Amapari, Laranjal do Jari in Amapá, Vitória do Jari, Gurupá and Breves in

Pará, in addition to some municipalities around the capital Macapá-AP, such as Chaves and Itaubal.

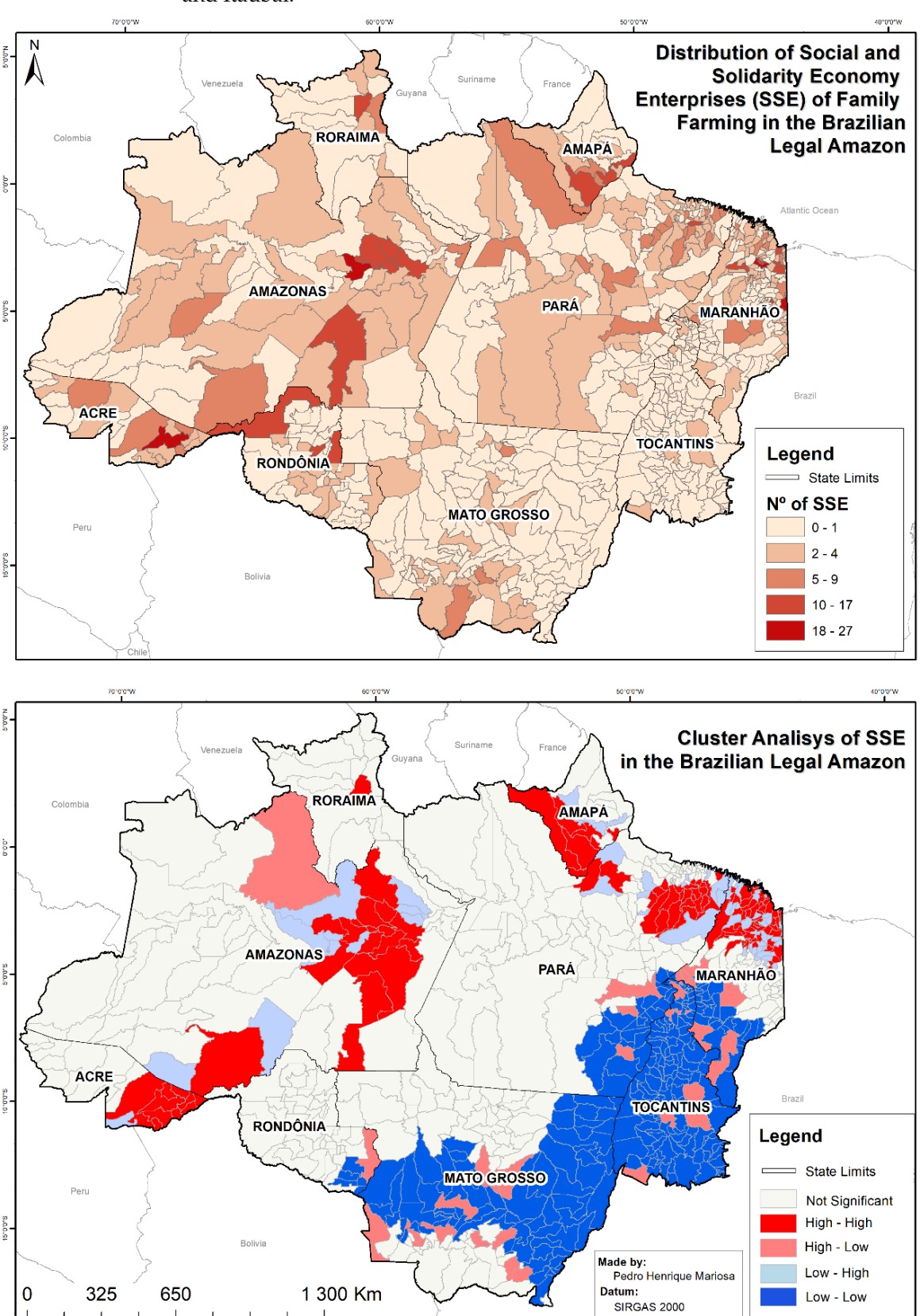

**Figure 4.** Distribution and cluster analysis of social and solidarity economy enterprises (SSE) of family farming in Brazilian Legal Amazon (*n* = 1298).

The fifth cluster was formed between the states of Pará and Maranhão, and this had a pivotal point in some municipalities on the border, such as Nova Espera do Piriá, Viseu and Paragominas in Pará and Câdino Mendes, Governador Nunes Freire, and Maracaçumé in Maranhão. This last cluster extended to the east to Itapecuru Mirim, Presidente Vargas, Presidente Juscelino, Santa Rita and Coroatá, close to Codó in Maranhão, a relevant municipality in terms of the number of SSE enterprises for family farming, and to the west to Igarapé-Miri and Limoeiro do Ajuru, near Cametá, in Pará, a municipality connecting clusters four and five.

Finally, the large region formed by the low–low cluster and high–low outlier integrated the southwest region of Maranhão, southeast of Pará, the state of Tocantins in its entirety, from the east to the west of Mato-Grosso, and the southeastern region of Rondônia. This extensive region had a low distribution pattern of family farming SSE enterprises. Therefore, it should become a priority for municipal, state, and federal governments to generate opportunities for the incorporation of techniques, models, and organization aiming at local and sustainable territorial development.

The west/north region of the Legal Amazon is currently in a vicious circle of development with low technological input, and needs to undergo an induction process (incentives) to overcome this condition [79]. This is the region where the alto–alto cluster predominates, which means that there are organizations in a defined spatial pattern, with the ability to mediate these incentives for the promotion of territorial transformations in their territories. The spatial autocorrelation in this high-high cluster, finally, shows a workable alternative for the inversion of this agricultural paradigm, but for this to happen there is a need to balance the issue of investment by the government and civil society organizations that supply support to associations and cooperatives.

## 4. Discussion

Considering the environmental, social, and economic dimensions of sustainability, the main question that guided this study was whether SSE enterprises, in the form of cooperatives, associations, and local collectives of family farming, are sufficiently developed in the Brazilian Legal Amazon, and, territorially, in what way. In response to this question, the authors looked to conduct an analysis of the spatial distribution of establishments and enterprises of family farming to find where, in the Legal Amazon, the principles, values, techniques, and goals of sustainable territorial development that characterize SSE were present, when compared to non-family farming establishments.

We verified the occurrence of two clustering patterns, characterized by their positive statistical significance and presented in the form of high–high and low–low clusters. Under the name of high–high clusters, the group of municipalities that had many non-family establishments stood out, forming polygons with other nearby municipalities that also had many non-family establishments. These areas, encompassing the edges and frontiers of the Brazilian Legal Amazon, coincided with the areas with the highest rate of mechanization where, since the beginning of the expansion of the agricultural frontier, there has been forceful pressure on the Amazon rainforest, suppressing, degrading, and fragmenting plant cover.

At the same time, areas with a low–low cluster presented a concentration of few establishments, and had a regular to high mechanization index, which leads to the assumption of large territorial extensions under the tutelage and domain of few owners, evidencing, thus, land concentration. Such a positive relationship between the advance of agricultural mechanization and the concentration of non-family farming establishments, especially in the new agricultural frontier, highlights the need for investment programs and the inclusion of local population contingents of family farming establishments as an essential measure to achieve sustainable development.

Likewise, family farming establishments formed autocorrelation patterns or clusters including: groups of municipalities with high development potential and the presence of family farming establishments; those with few municipalities in the peripheral area of the

cluster with a low concentration of these; those with low potential for the development of family farming establishments, with the presence of municipalities with a high concentration of these being sporadic; and those with territorial spaces outside of a pattern of significant distribution, positive or negative. This classification and hierarchy of spaces achieved by the study suggests priority environments for the implementation of public policies to strengthen and develop family farming as a strategy for achieving local and sustainable development.

Then, we analyzed the presence and contribution of family farming SSE enterprises in the formation of interaction and collaboration networks for local and sustainable territorial development. When applying Moran's Local Index to find spatial zones, territories, or spaces in which there was spatial autocorrelation for the analyzed categories, we found that the distribution was not territorially homogeneous. While in the Amazon region there were regions with greater influence of the principles, values, techniques, and models of action that are typical of SSE enterprises, there were others with a low influence of the models proposed by family farming SSE enterprises.

We reinforce here that in spaces where there was the presence of establishments, whether conventional family farming or purely capitalist establishments, there was a coincidence of areas with a greater occurrence of reduced vegetation cover and a greater negative environmental impact of agriculture. Thus, based on the results presented, it is possible to suggest the effective contribution of cooperative and/or associative family agriculture to the environmental conservation of the Amazon and the effective participation of the local population in the benefits that continued economic practice and appropriate public policies can offer.

## 5. Conclusions

In this study, we intended to show that the Brazilian Legal Amazon is a significant and paradigmatic locus for the debate about the involvement of different organizations, agents, and actors in the formulation of public policies and in the promotion of sustainability, with an emphasis on social and solidarity economy enterprises of family farming. Given its value and global relevance, the local impacts are even more serious when there are no consistent and coordinated ways of acting in the Amazon territory that consider, for example, the unprecedented opportunities offered there for the achievement of environmental conservation and the well-being of the planet, people, and different sectors of society, as listed in the Sustainable Development Goals provided for in the United Nations 2030 Agenda.

With the procedures outlined here, it was possible to see that, in general, in areas with more intense processes of mechanization and increases in population density and land concentration, there was also a reduction in enterprises aligned with the SSE principles, and even, many times, the removal of enterprises from these principles entirely.

Considering these findings, the hypothesis that social and solidarity economy enterprises (SSE) are important elements in achieving the sustainability of the Amazon proved to be feasible, because in addition to promoting the organization and consolidation of territorial systems of sustainable economic development, these enterprises are designed to provide opportunities for the socio-environmental and economic integration of the population living in this region.

The process of implementing and developing family farming SSE enterprises is, however, antagonistic to the current model of agricultural development based on commodity production. Both the SSE and family farming, in general, are alternatives to the hegemonic model of production and consumption. Family farming is a model of farming that conflicts with the national farming paradigm and, therefore, the entry and maintenance of this type of enterprise often ends up being suppressed in the face of the power of capital in the agricultural frontier, such as in the case of soybean cultivation in the Legal Amazon. In this sense, the SSE can be an alternative for the inversion of this agricultural paradigm, but for this to happen there is a need to balance the issue of investment by the public authorities.

The formulation of public policies for sustainable development involves the direct and effective participation of society and is a priority for municipal, state, and federal governments, and it also generates opportunities for the incorporation of techniques, models, and organization aimed at local and sustainable territorial development. It is the material conditions of existence and the legitimate interests of well-being and a sustainable quality of life that one wants to achieve using the driving forces that shape the norms, laws, directives, stimuli, and incentives inscribed in public policies.

What we proposed in this study is not conclusive or generalizable. Data from the Agriculture Census of the municipalities obtained from the series of municipal surveys by the IBGE, referring only to the year 2017, were used, in which information on the composition of the enterprises was available. Therefore, for the assessment of progress in the expansion or reduction of the representativeness of these enterprises and establishments, it will be necessary to analyze the representations of the years 2018, 2019, and the future. We therefore suggest that further studies are undertaken when these Agriculture Censuses are published. Finally, we recommend the production of new studies comparing the performance of the different forms of organization of agriculture in other countries and local territorial situations. These could consider whether cooperative and/or associative family farming would be the best option for achieving productive inclusion, a reduction in hunger, and the conservation of the environment.

**Author Contributions:** Conceptualization, P.H.M. and H.d.S.P.; Methodology, D.F.M. and O.M.F.; Project administration, P.H.M.; Supervision, H.d.S.P.; Validation, D.F.M.; Visualization, D.d.M.C.; Writing—original draft, P.H.M. and H.d.S.P.; Writing—review & editing, D.F.M., O.M.F., D.d.M.C. and S.C.D.B. All authors have read and agreed to the published version of the manuscript.

**Funding:** This article received financial support via Public Notice PAINTER 003/2020 from the public agency FAPEAM (Fundação de Amparo à Pesquisa do Estado do Amazonas) and is the result of work supported by the UFAM and public agency CAPES (Universidade Federal do Amazonas e Coordenação de Aperfeiçoamento de Pessoal de Nível Superior).

**Data Availability Statement:** The databases analyzed in this current study, as well as the generated dataset, are available from the corresponding author upon reasonable request. The official databases mentioned in the text can, however, be explored: https://dados.gov.br/, accessed on 21 August 2021, http://www.in.gov.br/ accessed on 12 January 2021, and https://www.ibge.gov.br accessed on 18 December 2021.

**Conflicts of Interest:** The authors declare no conflict of interest.

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
