# Peer review of "Family Farming and Social and Solidarity Economy Enterprises in the Amazon: Opportunities for Sustainable Development"

_sustainability, doi:10.3390/su141710855_

Round 1

Reviewer 1 Report

I think this is a good job. The topic and research work are very interesting. I just have some minor comments.

1. On the section of introduction, the academic contribution of this paper to the research topic should be elaborated more clearly. I suggest the authors to add more texts to showcase the academic contribution of this paper, in order to promote the significance of this paper better.

2. The authors should add more discussion about the results, what is the limitation of your research work? Future research directions may also be mentioned. I think it is important and missing here.

Author Response

Dear Referee 1,
We appreciate your contributions and generosity, and now we will respond point by point. We apologize for no longer attached to the amended file, as we are in the process of reviewing the MDPI's official English.
About your thoughts:

  1. On the section of introduction, the academic contribution of this paper to the research topic should be elaborated more clearly. I suggest the authors to add more texts to showcase the academic contribution of this paper, in order to promote the significance of this paper better.

Answer: As the proposal of the article is innovative (The main question that guided this study was whether the organizations of the Social and Solidarity Economy (ESS), in the form of cooperatives and associations of family farming, are sufficiently consolidated in the Brazilian Legal Amazon) no other publications were found, in addition to those already cited, at least considered relevant enough to be incorporated into the text. The articles evaluated for the consolidation of the research topic were screened through a systematic review of the literature on family farming and the social and solidarity economy in Latin America, already published by us at https://globaljournals.org/GJHSS_Volume22/1-Systematic -Review.pdf. We believe it is reasonable, at this time, not to cite this reference in the text of the article to ensure that we are not identified as authors of the manuscript in this evaluation process.

Therefore, we will incorporate this suggestion between lines 73 and 76, informing about this process and that this would justify the academic contribution of the study and its ability to serve as a basis for promoting public policies. If the referees have any suggestions for the incorporation of articles, we would appreciate the suggestions.

2. The authors should add more discussion about the results, what is the limitation of your research work? Future research directions may also be mentioned. I think it is important and missing here.

Answer: At the end of the article there is a comment about this process, between lines 756 and 762:

“What we proposed here is not conclusive and at all generalizable. Data from the Agriculture Census of the municipalities obtained from the series of municipal surveys by the IBGE, referring only to the year 2017, for which information on the composition of the enterprises is available. Therefore, for assessment of the movement on the expansion or reduction of the representativeness of these enterprises and establishments, it will be necessary to analyze the representations of the years 2018, 2019 and future. We therefore suggest that these studies continue when these Agriculture Census are published.”

In this way, as we agree with your first comment, we will also add to this final part that we recommend the production of new studies comparing the performance of the different forms of organization of agriculture in other countries and local territorial situations, even to know if, indeed, the family farming organized in a cooperative and/or associative manner would be the best option for achieving productive inclusion, reducing hunger and environmental conservation.

Regarding the discussion, we will reinforce that in spaces where the presence of Establishments, whether from Conventional Family Agriculture or those that are purely capitalist, there is a coincidence of areas with the highest occurrence of reduced vegetation cover and a greater negative environmental impact of agricultural activities. So, based on the results presented, it is possible to suggest the effective contribution of cooperative and/or associative Family Agriculture for the environmental conservation of the Amazon and the effective participation of the local population in the benefits that the continued economic practice and valued through appropriate Public Policies can offer.

Finally, regarding the writing in English, we will submit the manuscript for review by a native speaker.

We greatly appreciate and accept these two considerations.

Reviewer 2 Report

In my opinion, the theoretical framework on the social and solidarity economy should be broader and include other references that have analyzed the case of the SSE in Brazil.

Author Response

Dear Referee 2,

First, we express our satisfaction with the accuracy and generosity of your assessment.

About your suggestion:
"In my opinion, the theoretical framework on the social and solidarity economy should be broader and include other references that have analyzed the case of the SSE in Brazil."

Answer: As the proposal of the article is innovative (The main question that guided this study was whether the organizations of the Social and Solidarity Economy (ESS), in the form of cooperatives and associations of family farming, are sufficiently consolidated in the Brazilian Legal Amazon) no other publications were found, in addition to those already cited, at least considered relevant enough to be incorporated into the text. The articles evaluated for the consolidation of the research topic were screened through a systematic review of the literature on family farming and the social and solidarity economy in Latin America, already published by us at https://globaljournals.org/GJHSS_Volume22/1-Systematic -Review.pdf. We believe it is reasonable, at this time, not to cite this reference in the text of the article to ensure that we are not identified as authors of the manuscript in this evaluation process.

We appreciate and accept suggestion. Therefore, we will incorporate this suggestion between lines 73 and 76, informing about this process and that this would justify the academic contribution of the study and its ability to serve as a basis for promoting public policies. If the referees have any suggestions for the incorporation of articles, we would appreciate the suggestions.

Reviewer 3 Report

The paper is well structured and the results are valid. The database covered is vast and the data presented are of high importance.

Author Response

Dear Referee 3,

We appreciate the referee's evaluation and generosity in relation to our academic production. In any case, we are always willing to receive new criticisms.

Kind Regards!

Reviewer 4 Report

The paper deals with an interesting topic that could somehow support public policies for sustainable development.

There are some small suggestions that could however improve the structure of the work and its contents.

In the abstract we talk about a census referred to in 2017. Is there anything more updated? The data could have changed, even in relation to the advent of the pandemic in the last 2 years.

I suggest to insert the paragraph Background as a sub-paragraph of the introduction. It is suggested to modify for structural purposes.

The methods paragraph seems unpacked into several small sub-paragraphs. This makes the speech less fluid. If you decide to leave the sub-paragraphs, it is suggested to extend and integrate them. Line 277 shows a link that you can visit. It is also suggested to insert it in the appropriate place at the end of the paper provided by the editors of Sustainability with the name of: "Data availability statement" before the references, together with all the other aspects related to conflict of interest, contributions of the author, etc. ,they are absent.

It is advisable to integrate the references and to insert the various database links present in the bibliography within the aforementioned part

Author Response

Dear Referee 4,
I would first like to thank you immensely for the quality of your suggestion and your generosity in your evaluation.

Answering your questions point by point.

"In the abstract we talk about a census referred to in 2017. Is there anything more updated? The data could have changed, even in relation to the advent of the pandemic in the last 2 years."

ANSWER: Thank you for alerting us to the importance of communicating in the article that this is the last Census, officially produced by the Brazilian government, available for consultation. We will only know whether or not the data has changed after publication of the information that is still being collected and processed.

I suggest to insert the paragraph Background as a sub-paragraph of the introduction. It is suggested to modify for structural purposes.

ANSWER: We accepted the suggestion and reordered the Background as a subtopic of the introduction, also changing the numbering of the other topics.

"The methods paragraph seems unpacked into several small sub-paragraphs. This makes the speech less fluid. If you decide to leave the sub-paragraphs, it is suggested to extend and integrate them."

ANSWER: We appreciate and accept the suggestion and removed the numbering of the methodology's subtopics, but we kept the methodology's logical sequence.

"Line 277 shows a link that you can visit. It is also suggested to insert it in the appropriate place at the end of the paper provided by the editors of Sustainability with the name of: "Data availability statement" before the references, together with all the other aspects related to conflict of interest, contributions of the author, etc. ,they are absent."

ANSWER:

We accept the suggestion and create the topics:

Data availability

The databases analyzed in this current study, as well as the generated dataset, are available from the corresponding author upon reasonable request. The official databases mentioned in the text can, however, be explored in the: https://dados.gov.br/, http://www.in.gov.br/ and https://www.ibge.gov.br.

Acknowledgements

This article received financial support via Public Notice PAINTER 003/2020 from the public agency FAPEAM and is the result of work supported by the public agency CAPES.

Author information

Prof. Dr. Pedro Henrique Mariosa [email protected], BR, Federal University of Amazonas; Prof. PhD. Henrique dos Santos Pereira [email protected], BR, Federal University of Amazonas; Prof. Dr. Duarcides Ferreira Mariosa [email protected], BR, Pontifical Catholic University of Campinas; Prof. Dr.           Orandi Mina Falsarella [email protected], BR, Pontifical Catholic University of Campinas Prof. Dr. Diego de Melo Conti, [email protected], BR, Pontifical Catholic University of Campinas; Prof. Dr. Samuel Carvalho De Benedicto [email protected], BR, Pontifical Catholic University of Campinas.

Contributions

Conceptualization, Pedro Henrique Mariosa and Henrique dos Santos Pereira; Methodology, Duarcides Ferreira Mariosa and Orandi Mina Falsarella; Project administration, Pedro Henrique Mariosa; Supervision, Henrique dos Santos Pereira; Validation, Duarcides Ferreira Mariosa; Visualization, Diego de Melo Conti; Writing – original draft, Pedro Henrique Mariosa and Henrique dos Santos Pereira; Writing – review & editing, Duarcides Ferreira Mariosa, Orandi Mina Falsarella, Diego de Melo Conti and Samuel Carvalho De Benedicto.

Correspondence to Pedro Henrique Mariosa.

Consent for publication

All authors give their full consent for publication.

Conflicting interests

The authors declare that there are no conflicting interests.

Finally, we reiterate our satisfaction with all the suggestions presented, we hope to have answered them satisfactorily.

Kind Regards!
